# Married women's autonomy in modern contraceptive utilization in Kutaber district, Northeast Ethiopia

Getachew Amanu Bogale[1]☯*, Mastewal Arefaynie Temesgen[1]☯, Kemal Ahmed Seid[2]‡, Bantalem Amanu Bogale[3]‡¤a

1 Department of Reproductive Health, School of Public Health, Wollo University, Dessie, Ethiopia,
2 Department of Public Health Officer, School of Public Health, Wollo University, Dessie, Ethiopia,
3 Department of Adult Health Nursing, St. Paul's Hospital Millennium Medical College, Addis Ababa, Ethiopia

☯ These authors contributed equally to this work.
‡ KAS and BAB contributed equally to this work.
¤a School of Nursing and Rehabilitation, Shandong University, Jinan, China
* gech637@gmail.com

## Abstract

Married women's decision-making autonomy in modern contraceptive utilization is extremely important for better maternal and child health outcomes. Most studies in Ethiopia have not incorporated women's freedom of movement and control over finance when assessing married women's autonomy in modern contraceptive utilization. Previously conducted studies measured women's autonomy using different variables. This study aimed to measure married women's autonomy in contraceptive use in Kutaber district using better outcome measuring tool. A community-based cross-sectional study was conducted from July 10 to August 13, 2024. Interviews were conducted with 420 married women selected via a simple random sampling technique from the study population. Bivariable logistic regression was performed, and variables with p-values < 0.25 were included in the multivariable logistic regression, with a p-value < 0.05 considered statistically significant. Four hundred and eight (408) married women participated, resulting a response rate of 97%. About two-thirds (63.5%, 95% CI (58.3%-68.4%)) of the married women had autonomy in modern contraceptive utilization. Having household decision-making power (Adjusted Odd Ratio (AOR: 6.59, 95% Confidence Interval (CI (3.82,11.36))), being ≥18 years old at first marriage (AOR: 2.65, 95% CI (1.45,4.86)), having 3–4 live children (AOR: 3.74, 95% CI (1.82,7.67)), having ≥ 5 children (AOR: 10.78, 95% CI (3.60,32.31)), attending secondary school (AOR: 3.15, 95% CI (1.38,7.19)), and being in a marital union for 5–10 years (AOR: 2.90, 95% CI (1.20,6.98)) were significantly associated with married women's autonomy in modern contraceptive utilization. The prevalence of married women's autonomy in modern contraceptive utilization was high. Tackling early marriage and empowering women through adult education programs are recommended to improve married women's autonomy.

**Data availability statement:** All relevant data are within the paper and its Supporting Information files.

**Funding:** The author(s) received no specific funding for this work.

**Competing interests:** The authors have declared that no competing interests exist.

## Introduction

Women's decision-making autonomy in family planning refers to the capacity of women to decide independently, or with their husbands or others about family planning needs and choices [1]. For better maternal and child health outcomes, women's autonomy in reproductive health-care decision-making, including contraceptive use, is critical [2].

Globally, 164 million reproductive-age women have an unmet need for family planning, with 287,000 maternal deaths per year [3,4]. The Sub-Saharan Africa region (SSA) accounted for the greatest percentage of women with an unmet need for family planning (26%) and 202,000 maternal deaths, which represents 70.38% of global maternal deaths [3,4]. Similarly, nearly one in five women in Ethiopia (22%) had an unmet need for family planning, and only 41% used modern contraceptive methods. This situation contributes to a high total fertility rate of 4.6 children per woman, which in turn leads to a high maternal mortality ratio of 412 per 100,000 live births, neonatal mortality rate of 33, infant mortality rate of 47, and under-five mortality rate of 59 per 1000 live births [5,6].

The population of Africa is currently growing faster than that of any other major region and is expected to account for 21% of the global population by 2050, up from just 9% in 1950 [7]. The population growth rate of Ethiopia is estimated to be 2.5%, which is still a relatively high growth rate [8].

Autonomous women enjoy their sexual and reproductive health rights [9], and addressing unmet family planning needs could help avert approximately 7 million under-five child deaths and prevent 450,000 maternal deaths in 22 priority countries worldwide [10].

Throughout the world, women constitute a large proportion of the poor, underemployed, and socially and economically disadvantaged. There is widespread recognition that virtually no society provides women with equal status to men, and women generally have lower social status and autonomy than men do [11]. In certain nations, the husband and his parents are the primary decision-makers regarding the use of contraception by wives. In a rural communities where women's educational status is low and economic dependency is high, women's decision-making power concerning contraceptive use is limited [12]. Consequently, secret contraceptive use accounts for 6–20% of all contraceptive use in SSA [13]. However, this potentially exposes them to emotional or physical violence if it is discovered. When women possess greater decision-making power in the household over their reproductive health and rights, the health of the family is better protected, which contributes to the productive forces of the country as a whole [9,14]. Highly autonomous women are more likely to use maternal health care services [15], including modern contraception [16]. Contraceptive use is important for preventing fetal, neonatal, and under-five deaths; avoiding high-risk pregnancies; and reducing maternal mortality [17] by preventing unintended pregnancy and unsafe abortion [18].

Key barriers that perpetuate gender inequality and hinder women's empowerment include women's lack of safety and mobility; women's lack of resources and decision-making; limited access to information, education, and technology; as well as the excessive time burden and dual responsibilities that women often face [19].

The International Conference on Population and Development (ICPD) outlined a broad range of initiatives that governmental and nongovernmental organizations should undertake to support gender equality and justice, as well as women's empowerment [20]. Similarly, leaders from 193 countries came together and set 17 Sustainable Development Goals (SDGs) to be achieved by 2030. Among these goals, SDG-5 specifically targets the achievement of gender equality and the empowerment of all women and girls worldwide. Nations across the globe, including Ethiopia, are striving to achieve SDG- 5. However, according to the United Nations SDGs progress report; there are concerns about achieving this goal due to gross inequalities in work and wages, the prevalence of unpaid women's work, such as childcare and domestic tasks, and discrimination in decision-making [21].

Except for a few studies [2,22], most Ethiopian research does not consider women's freedom of movement and their financial control when assessing married women's autonomy in the use of modern contraceptives [23–27]. Only some studies included women who were non-users of modern contraceptives [2,27,28], while the majority did not address this group [23–26,29,30]. Based on this gap, we decided to include these women in our study. Additionally, this study incorporated variables related to women's freedom of movement and their financial control while assessing married women's autonomy in the utilization of modern contraceptives in the Kutaber district.

## Materials and methods

### Ethics statement

The study obtained ethical approval letter from the Ethical Review Committee of Wollo University College of Medicine and Health Sciences (Ref. No. CMHS/275/13/12). All participants were informed about the study's objectives and written informed consent was obtained from each respondent. Interviews were conducted in an isolated area away from household members, allowing women to freely express their feelings and ideas. Strict privacy measures were implemented to ensure the confidentiality of the data. Respondents' identifiers, such as names, were not requested or recorded during the interviews to protect data confidentiality. The study was conducted in accordance with the principles of the Helsinki Declaration, and respondents' right to refuse to answer a few or all of the questions was respected.

### Study setting, period, design and population

A community-based cross-sectional study was conducted from July 10 to August 13, 2024, in the Kutaber District, Northeast Ethiopia. The district is located 437 km away from Addis Ababa, the capital city of Ethiopia. The district has 23 kebeles (sub-districts) with a total population of 109,746 people, of whom 53,380 are females. Among them, 22,202 were women of reproductive age who were estimated to potentially use contraceptives. The socio-demographic characteristics of the 23 kebeles (sub-districts) do not vary significantly. Modern family planning is a free service available at all public health facilities, allowing all women to access it without charge.

All married women of reproductive age (15–49 years) living in the Kutaber district were the source population, and the study population consisted of all married women of reproductive age in the selected kebeles.

### Sample size determination and sampling procedure

For this particular study, the sample size was determined using a single population proportion formula, considering the following assumptions: a 95% confidence level, a 5% margin of error, and a 53.8% prevalence of married women's decision-making autonomy in modern contraceptive use in the Dawro Zone, southern Ethiopia [28]. Additionally, a 10% nonresponse rate was accounted for, resulting in a final sample size of 420.

Six kebeles (sub-districts) were chosen by a random sampling technique from a total of 23 kebeles in the district, which includes 1,817 married women of reproductive age. For each selected kebele, the total number of households with married women of reproductive age was obtained from the family folder of the community health information system available

at the local health post. The heads of these households were then listed to create a sampling frame. Following this, the sample size (420) was proportionally allocated to the selected kebeles.

Finally, a simple random sampling method was employed to select the participants to be included in the study from the sampling frame (Fig 1). The usual place of residence of the participants was identified in collaboration with local health extension workers and the health development army.

## Variable measurements and definitions

To measure the degree of married women's autonomy in modern contraceptive utilization, three groups of women were established: ever users, current users, and nonusers [2,27,28].

For **current users**, three questions were asked: who in your family made the final decision regarding your current contraceptive use, method choice, and source of contraceptives. The possible responses were: women independently, women and husbands jointly, women and others, husbands independently, or someone else in the household. A score of 1 was given to women who made decisions independently or jointly with their husband or someone else in the household. However, if the decision was made solely by the husband or another person in the household, the score was 0. All

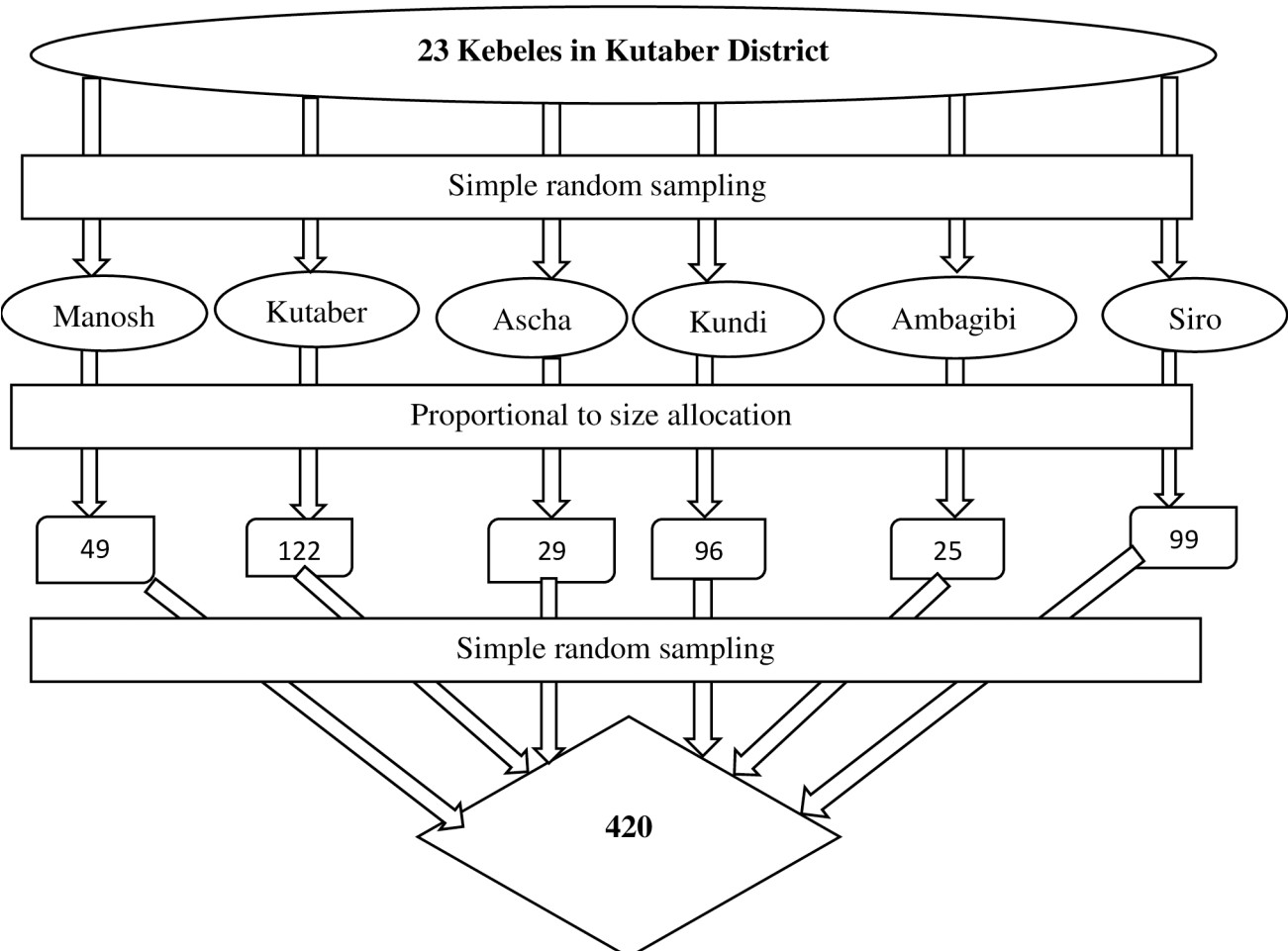

**Fig 1. Schematic presentation of sampling procedure for married women in Kutaber district, Northeast Ethiopia, 2024.**

participants who scored 1 on all three questions were categorized as "autonomous" and coded as 1, while those who scored 0 on any of the three questions were categorized as "not autonomous" and coded as 0 [23,24,29,30].

**Ever-user** women (but those currently not using modern contraceptives) were asked, "Who in your family made the final decision on stopping contraceptive use?" Women who stopped using modern contraceptives based solely on their own decision, or jointly with their husband or someone else in the household, were considered as "autonomous" and were coded as 1. Conversely, if the decision was made independently by her husband or someone else in the household, they were classified as "non autonomous" and coded as 0 [2].

For **non-users**, participants were asked, "Who in your family made a final decision not to use modern contraceptives?" Women who were not using modern contraceptives by their own decision, or jointly with their husband or someone else were, categorized as "autonomous" and coded as 1. In contrast, if the decision was made independently by her husband or another household member, they were considered "non autonomous" and coded as 0 [2]. Finally, married women's autonomy in contraceptive use among study units was established as a binary outcome variable by merging the three groups of women [2,27,28].

**Household decision-making power.** Refers to the ability of married women to make decisions within the household [22]. To measure the degree of women's household decision-making power, three questions were constructed based on items in the 2016 Ethiopian Demographic and Health Survey (EDHS 2016) [6]. "Who in your family usually has the final say on the following decisions?" These decisions include a woman's own health care, major household purchases, and visits to her family or relatives.

The possible responses were: women independently, women and husbands jointly, women and others, husbands independently, or someone else in the household. Women who made the decision independently, or in cooperation with their husband or another household member received a score of 1. Conversely, if the decision was made solely by the husband or someone else in the household, they received a score of 0. All participants who answered that they made decisions independently or jointly for all three items were categorized as "having household decision-making power" and coded as 1. The participants who reported that another person made decisions for any of the three items were categorized as "not having decision-making power" and coded as 0 [2,22,23,26,31–33].

**Freedom of movement.** A woman's ability to leave the house alone without asking permission from her husband. Freedom of movement consists of three items related to a woman's ability to leave the house without asking permission from her husband. These items assess whether she can visit her family or relatives, go to a health facility for her own health care service and go to the market. Each items has binary responses (yes or no). Those who answered 'yes' received a score of 1, while those who answered "no" received a score of 0. Those participants who answered "yes" for all three questions were categorized as "having freedom of movement" and coded as 1. The participants who answered "no" for any of the three items were categorized as "not having freedom of movement" and coded as 0 [2,22,34].

**Control over finance.** To measure the degree of married women's control over finance, the following four questions were used: whether they have separate bank account, if they save money for her future use, and whether they can decide how their earnings and their husband's earnings are spent. The questions for this variable were constructed from items in the EDHS 2016 [6]. The dummy response was scored in the same way as the freedom of movement item, while the remaining items were scored similarly to how household decision-making power was assessed. All participants who scored one on all four questions were considered to "have control over finance" and coded as 1. If not, they were considered to "have no control over finance" and coded as 0 [2,22,34].

The modern contraceptive methods: involve a product or medical procedures that interfere with the reproduction from sexual intercourse (oral contraceptive pills, injectable, implants, intrauterine devices, tubal ligation, and condoms).

**Knowledge of modern contraceptive methods.** Women were considered knowledgeable about modern contraceptive methods if they had heard of them and could name at least one modern female contraceptive method. Those who met these criteria were coded as 1, while those who did not were coded as 0 [5,28,29,35].

**Family structure.** A family consisting of only a couple and their unmarried children was considered as a nuclear family, whereas a family consisting of any individuals other than a couple and their unmarried children, excluding housemaids, was considered as an extended family structure [36].

**Media exposure.** Women who listened to the radio or watched television at least once a week were considered to have media exposure and were coded as 1; those who did not were coded as 0 [6,29].

**Types of marriage.** When a husband has another wife in addition to the respondent, the marriage is considered polygamous; if he does not, it is considered monogamous [6].

**Wealth index.** The wealth index of the household was computed using principal component analysis (PCA) after the data were collected. The variables were obtained from the 2016 EDHS and were coded as follows: 1 for the lowest, 2 for the second, 3 for the middle, 4 for the fourth, and 5 for the highest wealth quartiles [6].

## Data collection tools and procedure

The data were collected through face-to-face interviews using a structured questionnaire. The questionnaire was developed through a review of various studies and the EDHS 2016 [2,6,22–25,27,29]. The questionnaire was initially developed in English and then translated into the Amharic language, the local language of the study area. It was subsequently translated back into English to ensure its accuracy and consistency. During data collection, if respondents were unavailable or households were closed, up to three revisits were conducted before categorizing the case as a nonresponse.

## Data quality control

A pre-test was conducted on a similar population in the Ambasel District, involving 5% of the sample size (25 married women), prior to the actual data collection. Based on the findings of the pre-test, additional modifications and adjustments were made to the sequence and wording of the questionnaire. Cronbach's alpha was calculated and found to be 0.799, confirming the reliability of the instruments used. Daily supervision was carried out to ensure the completeness and accuracy of the data.

## Data processing and analysis

After the data were collected, they were coded, entered into Epi Data version 3.1, and then exported to the Statistical Package for Social Science version 25 (SPSS 25.0 Inc., Chicago, IL, USA) for data management and analysis. Descriptive statistics, such as frequencies and percentages were computed. The results are presented in text and tables. Since the outcome variable was dichotomous, a binary logistic regression model was used to identify the associated factors. The assumptions of logistic regression were checked using the Hosmer–Lemeshow goodness-of-fit test (P-value: 0.57). Multicollinearity among the independent variables was also assessed using the variance inflation factor (VIF), which was found to be less than 4.0. All variables with a p-value < 0.25 during bivariable logistic regression analysis were considered for multivariable logistic regression to control for possible confounders. In multivariable binary logistic regression, the strength of the statistical association was measured using an adjusted odds ratio (AOR) with 95% confidence intervals, and variables with a p-value <0.05 were considered statistically significant.

## Results

### Socio demographic characteristics

This study included 408 married women out of 420 eligible women, with a response rate of 97.14%. The mean age of the participants was 29.92 (±6.28) years. The average family size for the study population was 4.58 (±1.39). Most respondents (62.5%) were Muslims. Three hundred and two (74%) of the women were living in rural areas. The majority of the women, 323 (79.2%), resided in a nuclear family, whereas the remaining 85 (20.8%) lived in an extended family. Among

all the married women, 376 (92.2%) were in a monogamous marriages. Two hundred and forty-three (59.6%) of the married women had media exposure at least once a week (Table 1).

### Reproductive history of married women

The mean age at first marriage of married women was 18.92 (±2.81), with the minimum and maximum ages at first marriage being 14 and 30 years, respectively. Thirty-six percent (147) of the women were married before the age of 18 at their first marriage (Table 2).

### Modern contraceptive use history of married women

All the participants had heard about contraceptives and were aware of at least one method of family planning. The most well-known contraceptive method was the injectable method, with 391 (95.8%) women aware of it. Among the participants, 258 (63.9%) married women used modern contraceptive methods. Of these, 128 (49.6%) used injectable

**Table 1. Socio-demographic characteristics of married women in Kutaber district, Northeast Ethiopia, 2024.**

| Variables | Categories | Frequency | Percentage |
|---|---|---|---|
| **Age of women (years)** | 15–19 | 15 | 3.7 |
| | 20–24 | 70 | 17.2 |
| | 25–29 | 126 | 30.9 |
| | 30–34 | 96 | 23.5 |
| | 35–39 | 62 | 15.2 |
| | 40–44 | 32 | 7.8 |
| | 45–49 | 7 | 1.7 |
| **Women's educational status** | No formal education | 215 | 52.7 |
| | Primary school | 79 | 19.4 |
| | Secondary school | 67 | 16.4 |
| | Higher education | 47 | 11.5 |
| **Husbands' educational status** | No formal education | 166 | 40.7 |
| | Primary school | 106 | 26.0 |
| | Secondary school | 61 | 15.0 |
| | Higher education | 75 | 18.4 |
| **Duration of marital union** | <5 years | 127 | 31.1 |
| | 5–10`years | 147 | 36.1 |
| | >10 years | 134 | 32.8 |
| **Spousal age difference** | <5 years | 224 | 54.9 |
| | 5–9 years | 169 | 41.4 |
| | ≥10 years | 15 | 3.7 |
| **Women's occupation** | Housewife | 151 | 37.0 |
| | Farmer | 189 | 46.3 |
| | Merchant | 35 | 8.6 |
| | Gov't employer | 30 | 7.4 |
| | Private employer | 3 | 0.7 |
| **Husband's occupation** | Farmer | 232 | 56.9 |
| | Merchant | 113 | 27.7 |
| | Gov't employer | 53 | 13.0 |
| | Private employer | 10 | 2.5 |

**Table 2. Reproductive history of married women in Kutaber district, Northeast Ethiopia, 2024.**

| Variables | Categories | Frequency | Percentage |
|---|---|---|---|
| **Unintended pregnancy** | Yes | 21 | 5.1 |
| | No | 387 | 94.9 |
| **Abortion** | Yes | 56 | 13.7 |
| | No | 352 | 86.3 |
| **Number of Live Children** | No Children | 183 | 44.9 |
| | 1 to 2 Children | 32 | 7.8 |
| | 3-4 Children | 128 | 31.6 |
| | ≥ 5 Children | 65 | 15.7 |
| **Parity** | No Children | 171 | 41.9 |
| | 1 to 2 Children | 32 | 7.8 |
| | 3-4 Children | 128 | 31.4 |
| | ≥ 5 Children | 77 | 18.9 |
| **Age at first Marriage** | ≥ 18 Years old | 261 | 64 |
| | < 18 Years old | 147 | 36 |

contraceptive methods, followed by implants 109 (42.2%), intrauterine contraceptive devices (IUCDs) 18 (7.0%), and oral pills 3 (1.2%). Among the total current contraceptive users, 66 (25.6%) used contraceptive methods to limit pregnancies, 173 (67.05%) for spacing, and 19 (7.36%) to delay pregnancy. About 333 (81.6%) women reported that their husbands approved of contraceptive use (Table 3).

### Household decision-making power, freedom of movement, and control over finance among married women

Among the participants, nearly three-fourths (71.3%) of married women made decisions about their own healthcare services jointly with their husbands, and half (50%) of them had the freedom to visit health institutions for their own healthcare services without asking permission from their husbands. Similarly, 90.9% of respondents jointly decide how the money they earned is used.

Overall, 205 (50.2%) married women had household decision-making power, while 52 (12.7%) and 83 (20.3%) of the study participants had freedom of movement and control over financial resources, respectively (Table 4).

### Married women's autonomy in contraceptive utilization

The overall autonomy of married women in modern contraceptive utilization was 259 (63.5%). Among the current users, 178 (69.0%) of decisions to use modern contraceptives were made jointly, while about one in five (22.1%) decisions were made solely by the husband. The choice of contraceptive methods was mainly made by women alone, with 179 women (69.4%) making those decisions independently. Among those who stopped using contraceptives, nearly half (57.5%) did so based on their husband's decision alone (Fig 2).

### Factors associated with married women's autonomy in modern contraceptive utilization

Women's educational status, exposure to media, duration of marital union, age at first marriage, total number of live children, husband approval of family planning, household decision-making power, control over finance, and freedom of movement had p-value<0.25 in the bivariable analysis and were entered into the multivariable analysis. Finally, in the multivariable analysis, women's educational status, age at first marriage, total live children, household decision-making power, and duration of marital union remained statistically associated with married women's autonomy in contraceptive utilization.

**Table 3. Contraceptive history of married women in Kutaber District, Northeast Ethiopia, 2024.**

| Variables | Categories | Frequency | Percentage |
|---|---|---|---|
| **Method listed** | Injectable | 391 | 95.8 |
| | Oral pills | 274 | 67.2 |
| | Implanon | 378 | 92.6 |
| | Condom | 111 | 27.2 |
| | IUCD | 289 | 70.8 |
| | Tubal ligation | 17 | 4.2 |
| | Vasectomy | 1 | 0.2 |
| **Ever used modern contraceptive** | Yes | 404 | 99.0 |
| | No | 4 | 1.0 |
| **Currently use modern contraceptive** | Yes | 258 | 63.9 |
| | No | 146 | 36.1 |
| **Husband know the method used** | Yes | 241 | 93.4 |
| | No | 17 | 6.6 |
| **Reason for not using contraceptive currently** | Pregnant/amenorrhea | 67 | 44.7 |
| | Due health concern | 13 | 8.7 |
| | Wanted other children | 51 | 34 |
| | Religious matter | 3 | 2.0 |
| | Currently lactating | 3 | 2.0 |
| | Husband opposition | 12 | 8.0 |
| | No awareness on f/p | 1 | 0.7 |
| **Husband approval of contraceptive** | Yes | 333 | 81.6 |
| | No | 75 | 18.4 |
| **Source of information (Mostly)** | My friends | 15 | 3.7 |
| | Health post | 195 | 47.7 |
| | Health center | 187 | 45.8 |
| | Mass media | 11 | 2.7 |
| **Source of service (Mostly)** | Health post | 161 | 39.9 |
| | Health center | 243 | 60.1 |

In this study, women who had attended secondary school were 3.15 times (AOR: 3.15, 95% CI: 1.38,7.19) more likely to have autonomy in modern contraceptive use compared with those who had no formal education. Women with household decision-making power were 6.59 times (AOR: 6.59, 95% CI: 3.82-11.36) more likely to be autonomous in contraceptive utilization than those without such power. The study participants who married at the age 18 or older at first marriage were 2.65 times (AOR: 2.65, 95% CI: 1.45-4.86) more likely to excercise autonomy in contraceptive use.

Women who had lived 5–10 years with their current husbands were 2.90 times (AOR: 2.90, 95% CI: 1.20-6.98) more likely to be autonomous in contraceptive use than those who had lived together for fewer than five years. Similarly, women who had 3–4 children (AOR: 3.74, 95% CI: 1.82-7.67) and ≥ 5 children (AOR: 10.78, 95% CI: 3.60-32.31) were 3.74 and 10.78 times more autonomous in modern contraceptive use than women who had no children, respectively (Table 5).

## Discussion

The findings of this study showed that the overall prevalence of married women's decision-making autonomy in contraceptive utilization was 63.5% (95% CI: 58.3%–68.4%). This finding was consistent with studies conducted in Guinea (65.47%) [37], Mizan Aman (67.2%), and Gedeo zone (67.4%) in southern Ethiopia [25,29]. However, it was higher than

the results reported in studies conducted in Sub-Saharan African countries (25.28%), where the prevalence ranged from 9.9% in Burundi to 43.24% in Niger [30], as well as in East African countries (18.91%) [38], high fertility regions of Ethiopia (17.2%) [39], and a recent national study of Ethiopia (21.6%) [40]. Additionally, the prevalence was also higher than studies performed in Malawi (13%), Adwa town in northern Ethiopia (39.5%), Dawro Zone in southern Ethiopia (53.8%), Dinsho woreda (52%), and Dire Dawa city (55.2%) [2,23,27,28,41]. This discrepancy might be due to differences in outcome variable measurement, sample size, religion, culture, and timing of the studies. The number of questions used to measure women's decision-making autonomy in contraceptive utilization varied among these studies, which could contribute to the differing prevalence rates of women's autonomy [2,23,27,28,41].

Conversely, the finding was lower than those of studies conducted in Metekel Zone (71.0%) [26], Sekota in northwest Ethiopia (77.3%) [35], and Basoliben in northwest Ethiopia (80%) [24]. This difference may be attributed to variations in marital status, religion, and culture; for instance the study in Sekota included unmarried women. Married women often face significant cultural and partner-related constraints that limit their decision-making power regarding contraceptive use, while unmarried women typically do not encounter the same level of partner influence. This difference can lead to a higher likelihood of unmarried women making independent decisions in contraceptive use [42,43].

In this study, joint decision-making with husbands among current contraceptive users was 69% (95% CI: 63.6%–74.4%), which was consistent with the National 2016 EDHS report (73%) [6], and the study performed in the Gedeo zone (67.4%) [25]. However, this figure was higher than that reported in Adwa Town (53.4%) [2]. About one in five women (22.1% (95% CI: 17.1%-27.1%)) used contraceptives on the basis of their husbands' decisions alone, which was higher

**Table 4. Household decision-making power, freedom of movement and control over finance among married women in Kutaber district, Northeast Ethiopia, 2024.**

| Variables | Decisions | Who made decisions frequency (%) | | | |
|---|---|---|---|---|---|
| | | Women alone | Jointly | Husband alone | Other |
| **Household decision making power** | Her own healthcare | 51 (12.5) | 291 (71.3) | 62 (15.2) | 4 (1.0) |
| | Major household purchases | 0 (0.0) | 231 (56.6) | 161 (39.5) | 16 (3.9) |
| | Visits her family/relatives | 27 (6.6) | 225 (55.2) | 140 (34.3) | 16 (3.9) |
| **Freedom of movement** | | **Categories** | | **Frequency** | **Percent** |
| Without asking permission from her husband can she go out alone | To visit family | Yes | | 59 | 14.5 |
| | | No | | 349 | 85.5 |
| | To health facility for her own health care | Yes | | 204 | 50 |
| | | No | | 204 | 50 |
| | To local market | Yes | | 272 | 66.7 |
| | | No | | 136 | 33.3 |
| **Control over finance** | | **Categories** | | **Frequency** | **Percent** |
| Her own saving accounts | | Yes | | 100 | 24.5 |
| | | No | | 308 | 75.5 |
| Save money for future use | | Yes | | 134 | 32.8 |
| | | No | | 274 | 67.2 |
| Decisions how the money she earned used | | Women alone | | 25 | 6.1 |
| | | Jointly | | 371 | 90.9 |
| | | Husband alone | | 12 | 3.0 |
| Decisions how the money her husband earned used | | Women alone | | 8 | 1.9 |
| | | Jointly | | 265 | 65 |
| | | Husband alone | | 135 | 33.1 |

than the rates reported in the 2016 EDHS (5%) and Adwa Town, Northern Ethiopia (12.4%) [2,6]. The discrepancy arises from the fact that this study found nearly two-thirds (67%) of respondents had been in their marital unions for less than 10 years, compared to 53% in the 2016 EDHS. As the duration of marital unions increases, women's decision-making power regarding contraceptive use improves [9,44–47].

On the other hand, 57.5% (95% CI: 50%-65.8%) of married women stopped contraceptive use due to their husbands' decision alone, which was higher than the figures reported in the 2016 EDHS (10%) and in Adwa Town (28.2%) [2,6]. In general, among current contraceptive users, 77.9% of married women were involved in the decision-making process regarding contraceptive use. Similarly, among those who had stopped using contraceptives, 41.1% of married women were participated in the decision-making process to discontinue use. These figures were lower than those reported in the 2016 EDHS, which indicated that 95% and 88% were involved, respectively [6]. The possible explanation for this gap may be attributed to the fact that the national study examined multiple districts, as well as differences in culture, educational attainment, media exposure, and wealth index across the various settings.

The findings of this study revealed that women who attended secondary school were more likely to have autonomy in modern contraceptive utilization compared to those with no formal education. This findings is supported by previous studies [2,26,29,30,37,39]. Education empowers women to achieve independence and acquire essential knowledge for making informed decisions about their reproductive health. It also facilitates collaborative decision-making with their partners on health issues and social activities, such as visiting relatives, enabling them to share experiences and assert their reproductive and human rights [30,39,48].

Women who had household decision-making power were more likely to have autonomy in modern contraceptive utilization than those who did not have such power. This finding was congruent with studies conducted in Ethiopia, Zambia, and

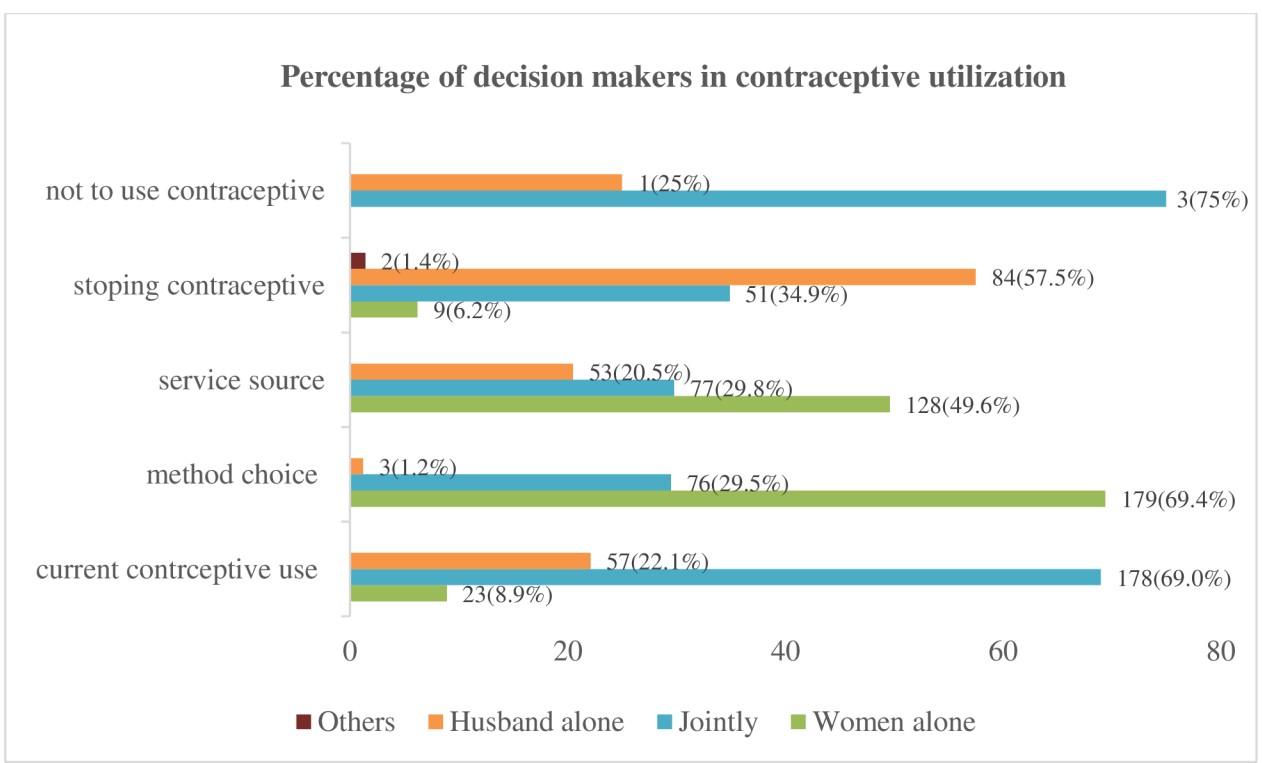

**Fig 2. Married women's autonomy in modern contraceptive utilization in Kutaber District, Northeast Ethiopia, 2024.**

Global Public Health

Iran [23,32,48,49]. In developing countries, most women hold inferior positions compared to their husbands in all aspects of decision-making; they often rely entirely on their male partners' decisions regarding contraception usage and reproductive life [50]. Empowered women play a significant role in reducing maternal and newborn morbidity and mortality by preventing unintended pregnancies and unsafe abortions. The household decision-making power of women can improve their contraceptive utilization, thereby achieving better maternal and child health outcomes [16,28].

Women who were married at the age of 18 years or older in their first marriage were more likely to be autonomous on contraceptive use than those who married under the age of 18. This finding is supported by a study done in Ethiopia

**Table 5. Factors associated with married women's autonomy in modern contraceptive utilization in Kutaber district, Northeast Ethiopia, 2024.**

| Variable | Autonomy in contraceptive utilization | | COR (95% CI) | AOR (95% CI) |
|---|---|---|---|---|
| | Yes | No | | |
| **Women's educational status** | | | | |
| No education | 118 | 97 | 1.0 | 1.0 |
| Primary school | 48 | 31 | 1.27(0.75–2.15) | 1.57(0.80–3.08) |
| Secondary school | 53 | 14 | 3.11(1.62–5.94)* | 3.15(1.38–7.19)* |
| Higher education | 40 | 7 | 4.69(2.01–10.95)* | 3.00(0.92–9.77 |
| **Total live children** | | | | |
| No children | 108 | 75 | 1.0 | 1.0 |
| 1-2 children | 26 | 6 | 3.00(1.82–7.66)* | 0.82(0.25–2.68) |
| 3-4 children | 86 | 43 | 1.39(0.86–2.22) | 3.74(1.82–7.67)** |
| ≥ 5 children | 39 | 25 | 1.08(0.60–1.93) | 10.78(3.60–32.31)** |
| **Marital union duration** | | | | |
| >10 years | 82 | 52 | 1.16(0.77–1.91) | 1.40(0.66–2.98) |
| 5-10 years | 104 | 43 | 1.78(1.08–2.93)* | 2.90(1.20–6.98)* |
| <5 years | 73 | 54 | 1.0 | 1.0 |
| **Age at first marriage** | | | | |
| ≥ 18 years old | 187 | 74 | 2.63(1.72–4.00)** | 2.65(1.45–4.86)* |
| < 18 years old | 72 | 75 | 1.0 | 1.0 |
| **Media exposure** | | | | |
| Yes | 175 | 68 | 2.48(1.64–3.75)** | 1.27(0.73–2.20) |
| No | 84 | 81 | 1.0 | 1.0 |
| **Husband approval of family planning** | | | | |
| Yes | 219 | 114 | 1.68(1.01–2.79)* | 0.97(0.52–1.80) |
| No | 40 | 35 | 1.0 | 1.0 |
| **Household decision-making power** | | | | |
| Yes | 173 | 32 | 7.35(4.60–11.75)** | 6.59(3.82–11.36)** |
| No | 86 | 117 | 1.0 | 1.0 |
| **Freedom of movement** | | | | |
| Yes | 44 | 8 | 3.60(1.64–7.89)** | 2.12(0.85–5.26) |
| No | 215 | 141 | 1.0 | 1.0 |
| **Control over finance** | | | | |
| Yes | 67 | 16 | 2.90(1.61–5.22)** | 0.75(0.33–1.67) |
| No | 192 | 133 | 1.0 | 1.0 |

**Statistically significant at p-value <0.001, *statistically significant at p-value <0.05,

1.0 Reference category, COR Crude odd ratio, AOR Adjusted odd ratio, CI Confidence interval.

[39,40]. Early marriage is widely recognized as a harmful customary practice that undermines women's full enjoyment of their economic and social rights. It negatively impacts their sexual and reproductive health, denies them opportunities for education, autonomy, and decision-making, and exposes them to gender-based domestic violence. Conversely, women who married after the age of 18 typically have better access to education, which may empower them in the decision-making processes related to their reproductive health, including contraceptive utilization [39,40].

Compared with their counterparts, women who had lived with their current husbands for 5–10 years were more likely to be autonomous on contraceptive utilization. This finding was supported by studies conducted in various regions of Ethiopia, South Africa, and 33 countries in SSA, utilizing recent demographic and health survey datasets [9,44–47]. This might be because when a couple lives together for a longer period, women may reduce their fear of their husbands, feel more confident, and develop a habit of interspousal communication or discussion about modern contraceptive needs and choices [47].

Study participants who had more living children were more autonomous in their use of modern contraceptives than those who had no children, as supported by various previous studies [25,30,32,45]. When couples have more living children in the household, the decision-making autonomy of women improves. This improvement can be attributed to the educational opportunities available to their children, which in turn influences parents' attitudes toward seeking reproductive health services [38].

## Implications of the study

Programs and public health campaigns should take into account the cultural and religious contexts that affect women's autonomy in contraceptive use. By customizing messages to align with local beliefs, acceptance and use of family planning methods can be improved, ultimately supporting the attainment of Sustainable Development Goals 3 and 5 by 2030.

Policies should promote collaborative decision-making among couples about family planning, which can be enhanced through community programs that foster discussions on reproductive health within families. Additionally, reinforcing laws against early marriage can enhance women's autonomy. Policymakers should support establishing the legal marriage age at 18 or older, ensuring women have improved educational and economic prospects.

Creating support groups for women can offer a space to share experiences and challenges regarding contraceptive use, promoting a sense of community and empowerment. It is crucial to continuously monitor women's autonomy in using contraceptives and the effectiveness of the programs in place. Furthermore, feedback systems should be implemented to adjust strategies according to the needs and outcomes of the community.

## Limitations of the study

Women's autonomy was measured by self-reports, which are likely to be subject to social desirability bias. Qualitative data were not collected to support the findings and husbands were not included in the study. The cross-sectional nature of the study did not allow for the establishment of a causal relationship between demographic and other characteristics of study participants and decision-making autonomy in modern contraceptive utilization. This study did not take into account the role of social media in the media exposure status of married women. We also focused only on modern contraceptive method use and did not fully report on traditional contraceptive methods. While traditional contraceptive methods play a role in family planning, their limitations in effectiveness and reliability warrant a focus on modern alternatives.

## Conclusion

Overall, married women's autonomy in the utilization of modern contraceptives was relatively high compared to the recent national study in Ethiopian using the EDHS 2016 data. Women's educational status, household decision-making power, age at first marriage being ≥18 years old, having more living children, and being in a marital union for 5–10 years were positively associated with married women's autonomy in modern contraceptive utilization. Future researchers should

include stakeholders in modern contraceptive use: health care providers to ensure fidelity in the use of modern contraceptives; and husbands to promote their involvement in modern contraceptive use.

## Acknowledgments

We are grateful to all the study participants for their voluntary participation in the interview process. We also want to acknowledge the Kutaber District Health Office for accessing the community health information system at the health post level and providing health extension workers' guidance for this study.

## Author contributions

**Conceptualization:** Getachew Amanu Bogale, Mastewal Arefaynie Temesgen.

**Data curation:** Getachew Amanu Bogale, Mastewal Arefaynie Temesgen, Kemal Ahmed Seid, Bantalem Amanu Bogale.

**Formal analysis:** Getachew Amanu Bogale, Mastewal Arefaynie Temesgen.

**Investigation:** Getachew Amanu Bogale, Mastewal Arefaynie Temesgen, Kemal Ahmed Seid, Bantalem Amanu Bogale.

**Methodology:** Getachew Amanu Bogale, Mastewal Arefaynie Temesgen.

**Software:** Getachew Amanu Bogale, Mastewal Arefaynie Temesgen, Kemal Ahmed Seid, Bantalem Amanu Bogale.

**Validation:** Getachew Amanu Bogale, Mastewal Arefaynie Temesgen.

**Visualization:** Kemal Ahmed Seid, Bantalem Amanu Bogale.

**Writing – original draft:** Getachew Amanu Bogale, Mastewal Arefaynie Temesgen, Kemal Ahmed Seid, Bantalem Amanu Bogale.

**Writing – review & editing:** Getachew Amanu Bogale, Mastewal Arefaynie Temesgen, Kemal Ahmed Seid, Bantalem Amanu Bogale.

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
