## [Decision Letter · Decision Letter 0]

6 Dec 2024

PGPH-D-24-02291

Married women’s autonomy in modern contraceptive utilization in Kutaber district, Northeast Ethiopia

Dear Dr. Bogale,

Thank you for submitting your manuscript to PLOS Global Public Health. After careful consideration, we feel that it has merit but does not fully meet PLOS Global Public Health’s publication criteria as it currently stands. Therefore, we invite you to submit a revised version of the manuscript that addresses the points raised during the review process.

Dear Authors,

Thank you for submitting your manuscript, “Married women’s autonomy in modern contraceptive utilization in Kutaber district, Northeast Ethiopia.” This is an important and timely study that contributes to our understanding of women’s reproductive health autonomy, particularly in the context of modern contraceptive use in Ethiopia. However, after a thorough review of the manuscript, several key revisions are necessary to improve clarity, rigor, and overall impact. Below, I provide detailed feedback on various aspects of your study, which should be addressed in the revised version.

First, I recommend placing your findings within a broader context by comparing them with similar studies from Sub-Saharan Africa and other global contexts. Additionally, clearer definitions and consistent usage of terms like autonomy, decision-making power, and empowerment are necessary to avoid ambiguity.

In terms of methodology, more detail is needed regarding the development and validation of the measures for decision-making power, freedom of movement, and financial control. How were these constructs operationalized, and were they pre-tested? Additionally, the process of data collection from illiterate participants and how privacy was maintained should be clarified. Discussing checks for multicollinearity and model fit in your regression analysis would further enhance the validity of your results.

In the discussion, a more nuanced interpretation of the discrepancies in contraceptive autonomy compared to EDHS data is needed, with an emphasis on potential cultural, religious, or regional factors. The section on empowering women could also benefit from more concrete suggestions for interventions or policies. I also suggest adding a section on the broader implications of your study for global public health and policy.

A careful review of the manuscript’s English language and grammar is recommended. Finally, the limitations section should address the potential underreporting of non-modern contraceptive use and clarify the rationale for focusing on modern methods. In the conclusion, specify the baseline against which autonomy is measured to provide clearer context.

Kindest regards

Please ensure that your decision is justified on PLOS Global Public Health’s publication criteria  and not, for example, on novelty or perceived impact.

We look forward to receiving your revised manuscript.

Kind regards,

Esra Keles

Academic Editor

Journal Requirements:

1. You indicated that you had ethical approval for your study. In your Methods section, please ensure you have also stated whether you obtained consent from parents or guardians of the minors included in the study or whether the research ethics committee or IRB specifically waived the need for their consent.

Additional Editor Comments (if provided):

Reviewers' comments:

Reviewer's Responses to Questions

**Comments to the Author**

1. Does this manuscript meet PLOS Global Public Health’s publication criteria ? Is the manuscript technically sound, and do the data support the conclusions? The manuscript must describe methodologically and ethically rigorous research with conclusions that are appropriately drawn based on the data presented.

Reviewer #1: Yes

Reviewer #2: Yes

Reviewer #3: Yes

2. Has the statistical analysis been performed appropriately and rigorously?

Reviewer #1: Yes

Reviewer #2: Yes

Reviewer #3: Yes

3. Have the authors made all data underlying the findings in their manuscript fully available (please refer to the Data Availability Statement at the start of the manuscript PDF file)?

Reviewer #1: Yes

Reviewer #2: Yes

Reviewer #3: Yes

4. Is the manuscript presented in an intelligible fashion and written in standard English?

Reviewer #1: Yes

Reviewer #2: Yes

Reviewer #3: Yes

5. Review Comments to the Author

Reviewer #1: The authors stated that all modern family planning services are provided to the clients for free at public health facilities (lines 109-110); if so, studying many variables related to women's autonomy on money or income might not be relevant because they do not go with the aims of the study. It is better to use standard scoring tools rather than citing previous studies to measure some parameters such as control over finance and knowledge of modern contraceptive methods (Lines 184-195). In the data collection tool, it is stated that the authors used questionnaires as a tool. However, in the result part, 52% of the participants did not have formal education (could not read and write). How were their data obtained? please justify it. Regarding privacy, it is better to justify the mechanisms used to maintain the privacy of the study participants during data collection. What scientific standards were used to make categories of the number of children the study participants had (1-2, 3-4 or >5, etc)? please justify.

Reviewer #2: Introduction

-Terms like “LARC” (long-acting reversible contraceptives) and the full scope of “women’s autonomy” are used without early clarification.Terms like "autonomy," "decision-making power," and "empowerment" are used somewhat interchangeably. Define autonomy clearly as it relates to family planning, and use consistent language to discuss this concept throughout the introduction to avoid confusion.

- The introduction touches on autonomy in a general sense but could benefit from a specific discussion of gender norms, family structures, and barriers unique to Ethiopia. This might include cultural, religious, or regional differences that shape reproductive health decision-making. Exploring recent data on how these dynamics influence contraceptive use specifically in Ethiopia would make the setting more tangible and relevant.

-lines 53-59 provide overlapping statistics on maternal mortality, neonatal mortality, and fertility rates. Summarizing or consolidating some of this information can help keep the reader focused on the core issues.

Some phrases are repeated or redundant. For example, lines 77-80 describe the benefits of autonomy in similar ways to prior sentences. Condensing these could improve flow and readability.

Method

-Selecting six kebeles from 23 using random sampling ensures representation across different sub-districts, but it is essential to specify if each kebele has a similar or varying demographic profile. If demographics vary greatly, stratifying before sampling might have ensured a more balanced representation.

-Allocating the sample size proportionally to each kebele is appropriate and helps maintain representativeness. However, it is unclear how households with eligible women were listed or verified, which may raise concerns about accuracy in the sampling frame.

-Include a citation to the Statistical Package for Social Science (SPSS) version 25 for data management

-Translating the questionnaire into Amharic is excellent for local understanding, but it would be useful to mention if any back-translation process was conducted to ensure accuracy.

-The study mentions that adjustments were made based on the pre-test findings but does not specify what modifications were made. More transparency on the exact changes (e.g., specific wording modifications or sequence adjustments) would help readers understand how the tool was refined. Additionally, it would be helpful to describe how the pre-test sample was selected and whether it differed from the actual study population (e.g., in terms of socio-demographic characteristics). Furthermore, it would be valuable to report the Cronbach's alpha result to indicate the tool's reliability.

-The use of multivariable logistic regression to control for confounding is appropriate. However, the study does not mention any checks for multicollinearity between independent variables, which could affect the validity of the regression results. Including a statement on how multicollinearity was addressed (e.g., using variance inflation factors) would improve the analysis section.

_While the study uses the Hosmer-Lemeshow test to check model fit, it does not mention the goodness-of-fit results or how they influenced model selection.

-Describe the specific steps taken to protect data confidentiality, such as how data were stored and whether any identifiers were removed

Reviewer #3: The review has been uploaded as an attachment.

This study considered autonomy in use of modern contraceptives among married Ethiopian women in a Kutaber district of Ethiopia. Authors have done a great job with the study.

Authors should consider placing findings from this study in context of findings not only from Ethiopia but also other countries in sub-Saharan Africa and countries like Ethiopia, globally. Authors need to provide strong justification for why this study is necessary and the particular gaps that this study is filling.

6. PLOS authors have the option to publish the peer review history of their article (what does this mean? ). If published, this will include your full peer review and any attached files.

**Do you want your identity to be public for this peer review?** For information about this choice, including consent withdrawal, please see our Privacy Policy .

Reviewer #1: No

Reviewer #2: **Yes: ** KHADIJAT ADELEYE

Reviewer #3: No

---

## [Decision Letter · Decision Letter 1]

18 Feb 2025

PGPH-D-24-02291R1

Married women’s autonomy in modern contraceptive utilization in Kutaber district, Northeast Ethiopia

Dear Dr. Bogale,

Thank you for submitting your manuscript to PLOS Global Public Health. After careful consideration, we feel that it has merit but does not fully meet PLOS Global Public Health’s publication criteria as it currently stands. Therefore, we invite you to submit a revised version of the manuscript that addresses the points raised during the review process.

Your revised manuscript has been assessed by two of the previous reviewers. Reviewer 3 requests a deeper comparison with existing surveys (see the reviewers' comments below).

In addition, I have some major concerns about the scoring and coding of many of the variables.

You have three groups of participants: current/ever/non-users of contraception. The autonomy score for current users is calculated differently to the scores for ever and non users. For current users there are three questions, each scored from 0-2 for a total score range of 0-6. However, for ever and non users there is a single question, scored 0 or 1. For current users, you split them at the mean score into "autonomous" and "not autonomous". However, we do not know what the mean was, and what level of autonomy is represented by the mean score.

For ever/non-users, a score of 1 means that decisions are made independently by the women or jointly with their husbands, whereas 0 means the decisions are made by someone else. But for current users, you have split this first category (independent or joint) into two (1=joint, 2=independent).

For current users to be scored  in an equivalent to past/non-users, "autonomous" would be defined as women who were independent or joint decision makers: This would mean a score of 1 or 2 on each question, for a total score of 3-6. However, a score of 3 may or may not reflect autonomy as this score would be given to any women who answers "joint" for all three questions (autonomous) or a women who answers 0, 1, and 2 across the three questions (not autonomous).

The issue here is that there may not be equivalence between current users categorised as "autonomous" and past/non users. If the majority of current users scored 0 or 1 on each of the three questions, the mean could be very low. As you used the mean score to split the current users into two groups, it could be the case that the autonomous group includes women whose decisions regarding contraceptive use were made by others. Or, if all participants answered 1 or 2 to all three questions, then the entire sample could be regarded as having autonomy, yet half of them would be categorised as having no autonomy. It would make more sense to define what set of scores indicates autonomy (e.g., a score of at least one on all three questions).

The coding of household decision making power has the same issue. What was the mean, and is it the case the categorisation accurately divides the participants into groups who do or do not have decision-making power?

The same concerns applies to the freedom of movement and control over finance variables.

As a result of the way the variables were scored and coded, the results tell us whether (for example) women who score above the mean on household decision-making power have higher odds of scoring above the mean on contraceptive autonomy than women who score below the mean on household decision-making power, *not* whether women who have household decision-making power have higher odds of having contraceptive autonomy than women without household decision-making power.

The 2016 EDHS defines participation in major household decisions as:

"Women are considered to participate in household decisions if they make decisions alone or jointly with their husband in all three of the following areas:

(1) the woman’s own health care,

(2) major household purchases, and

(3) visits to the woman’s family or relatives."

It is therefore essential that you revise your coding and analyses so that the categories for each of the variables more accurately reflects the meaning of that category. For example, household decision-making power should be coded as all participants who answer that they make decisions independently or jointly for all of the items relating to that variable. Any participant who reports that another person makes decisions for any of the items would be categorised as not having decision-making power.

Similar definition-based categories are needed for current user autonomy in modern contraceptive utilization, freedom of movement and control over finance. 

Could you please revise the manuscript to carefully address the concerns raised?

We look forward to receiving your revised manuscript.

Kind regards,

Steve Zimmerman, PhD

PLOS Staff Editor

Additional Editor Comments (if provided):

Reviewers' comments:

Reviewer's Responses to Questions

**Comments to the Author**

1. If the authors have adequately addressed your comments raised in a previous round of review and you feel that this manuscript is now acceptable for publication, you may indicate that here to bypass the “Comments to the Author” section, enter your conflict of interest statement in the “Confidential to Editor” section, and submit your "Accept" recommendation.

Reviewer #1: All comments have been addressed

Reviewer #3: (No Response)

2. Does this manuscript meet PLOS Global Public Health’s publication criteria ? Is the manuscript technically sound, and do the data support the conclusions? The manuscript must describe methodologically and ethically rigorous research with conclusions that are appropriately drawn based on the data presented.

Reviewer #1: Partly

Reviewer #3: Yes

3. Has the statistical analysis been performed appropriately and rigorously?

Reviewer #1: Yes

Reviewer #3: Yes

4. Have the authors made all data underlying the findings in their manuscript fully available (please refer to the Data Availability Statement at the start of the manuscript PDF file)?

Reviewer #1: Yes

Reviewer #3: Yes

5. Is the manuscript presented in an intelligible fashion and written in standard English?

Reviewer #1: Yes

Reviewer #3: Yes

6. Review Comments to the Author

Reviewer #1: It is revised good. Corrections are made.

Reviewer #3: Thank you for the opportunity to review this article.

The article presents a well-structured and methodologically sound study on the autonomy of married women in using modern contraceptives in Kutaber District, Ethiopia. The authors highlight an important issue concerning reproductive health and gender equality, and their findings provide valuable insights into the factors influencing women’s decision-making power in contraceptive use.

The authors have done a great job addressing previous concerns.

The study mentions discrepancies between its findings and Ethiopian Demographic and Health Survey (EDHS) 2016 data but does not fully explore potential explanations for these differences. A deeper comparison with more recent national surveys (e.g., the 2019 Mini DHS) could strengthen the discussion.

7. PLOS authors have the option to publish the peer review history of their article (what does this mean? ). If published, this will include your full peer review and any attached files.

**Do you want your identity to be public for this peer review?** For information about this choice, including consent withdrawal, please see our Privacy Policy .

Reviewer #1: **Yes: ** Kumlachew Mergiaw Abtew (PhD)

Reviewer #3: No

---

## [Decision Letter · Decision Letter 2]

1 May 2025

Married women’s autonomy in modern contraceptive utilization in Kutaber district, Northeast Ethiopia

PGPH-D-24-02291R2

Dear Mr. Bogale,

We are pleased to inform you that your manuscript 'Married women’s autonomy in modern contraceptive utilization in Kutaber district, Northeast Ethiopia' has been provisionally accepted for publication in PLOS Global Public Health.

Best regards,

Julia Robinson

Executive Editor

Reviewer Comments (if any, and for reference):

Reviewer's Responses to Questions

**Comments to the Author**

1. If the authors have adequately addressed your comments raised in a previous round of review and you feel that this manuscript is now acceptable for publication, you may indicate that here to bypass the “Comments to the Author” section, enter your conflict of interest statement in the “Confidential to Editor” section, and submit your "Accept" recommendation.

Reviewer #1: All comments have been addressed

Reviewer #4: All comments have been addressed

2. Does this manuscript meet PLOS Global Public Health’s publication criteria ? Is the manuscript technically sound, and do the data support the conclusions? The manuscript must describe methodologically and ethically rigorous research with conclusions that are appropriately drawn based on the data presented.

Reviewer #1: Partly

Reviewer #4: (No Response)

3. Has the statistical analysis been performed appropriately and rigorously?

Reviewer #1: Yes

Reviewer #4: (No Response)

4. Have the authors made all data underlying the findings in their manuscript fully available (please refer to the Data Availability Statement at the start of the manuscript PDF file)?

Reviewer #1: Yes

Reviewer #4: (No Response)

5. Is the manuscript presented in an intelligible fashion and written in standard English?

Reviewer #1: (No Response)

Reviewer #4: (No Response)

6. Review Comments to the Author

Reviewer #1: Still minor comments are not addressed.

Reviewer #4: The paper is well written statistically. The single population sample size with margin of error calculation makes sense. The final overall sample size with dropouts appears reasonable. The investigators are to be commended for the section on ‘Variable measurements and definitions', which was helpful.

The analysis is simple and appropriate both univariately and the multivariate approach (non adjusted and adjusted odds ratios) for this particular endeavor. The conclusions appear to follow from the analysis’s results. The authors noted relevant limitations which did not appear to impact the results overall.

On a minor note this reviewer is curious as to the possible reason as to why the wide (or appears to be wide) confidence interval on the adjusted odds ratio in Table 5 for the >= 5 children variable , AOR= 10.78, CI= (3.60-32.31).

7. PLOS authors have the option to publish the peer review history of their article (what does this mean? ). If published, this will include your full peer review and any attached files.

**Do you want your identity to be public for this peer review?** For information about this choice, including consent withdrawal, please see our Privacy Policy .

Reviewer #1: **Yes: ** Kumlachew Mergiaw Abtew

Reviewer #4: No
